# The Impact of Information on Attitudes toward Sustainable Wildlife Utilization and Management: A Survey of the Chinese Public

**DOI:** 10.3390/ani11092640

**Published:** 2021-09-08

**Authors:** Zhifan Song, Qiang Wang, Zhen Miao, Kirsten Conrad, Wei Zhang, Xuehong Zhou, Douglas C. MacMillan

**Affiliations:** 1College of Wildlife and Protected Area, Northeast Forestry University, Harbin 150040, China; 13136751029@163.com (Z.S.); miaozhen43566@163.com (Z.M.); 2Key Laboratory of Wetland Ecology and Environment, Northeast Institute of Geography and Agroecology, Chinese Academy of Sciences, Changchun 130102, China; qwang@neigae.ac.cn; 3International Association for Wildlife, Gungahlin, Canberra, ACT 2912, Australia; Kirsten@iaforwildlife.org; 4Durrell Institute of Conservation and Ecology (DICE), University of Kent, Canterbury CT2 7NR, UK; dcm@kent.ac.uk

**Keywords:** wild animals, wildlife conservation, sustainable utilization, information, awareness, agreement, China

## Abstract

**Simple Summary:**

The widespread dissemination of information related to wildlife utilization in new online media and traditional media undoubtedly impacts societal conservation concepts and attitudes, thus triggering public discussions on the relationship between conservation and utilization. In this study, questionnaires were distributed in seven major geographic regions of Chinese mainland to investigate the public’s awareness and agreement with information related to the utilization of wildlife in order to measure the impact of information on various issues relating wildlife conservation and utilization. The Chinese public had the greatest awareness and agreement with information that prevents unsustainable and illegal utilization, and the least awareness and agreement with information that promotes unsustainable utilization. It is also noteworthy that the Chinese public have higher levels of awareness and agreement with information that does not support utilization than with information that supports sustainable utilization. From our research, we can conclude that overall the public tends to support and be informed by a purist view of conservation, and there is significantly less support for conservation based on sustainable utilization. On this basis, we suggest that in the future, conservation education should seek to balance the public’s respect and love for nature which is often inspired by social media, with more scientific information about scientific understandings that influence conservation policy and practice.

**Abstract:**

The widespread dissemination of information related to wildlife utilization in new online media and traditional media undoubtedly impacts societal conservation concepts and attitudes, thus triggering public discussions on the relationship between conservation and utilization. A study on how public attitudes and concepts are affected by the related information on wildlife utilization is helpful to implement the scientific wildlife conservation and management strategies, and to propose targeted measures to optimize the information environment. We designed the questionnaire to investigate the public’s awareness and agreement with related information on wildlife utilization so as to measure how information with different dissemination channels, source types, and content orientation influenced the public’s concept of wildlife conservation and utilization. The questionnaire was distributed in seven major geographical regions throughout China. Out of a total of 1645 questionnaires that were collected, 1294 questionnaires were valid, with an effective rate of 78.7%. Results show that respondents had the greatest awareness of information on preventing unsustainable and illegal utilization, and the lowest awareness of information on promoting unsustainable utilization, and that awareness of information that against utilization was higher than that of information which supported sustainable utilization. At the same time, respondents showed the greatest agreement for information on preventing unsustainable utilization and the lowest agreement for information on promoting unsustainable utilization; also, their agreement with information that against utilization was higher than that for information which supported sustainable use. Respondents had a high level of awareness of information on wildlife related to COVID-19 provided by experts. Gender, age, the level of development of the city in which they live, education, vegetarianism, and religious beliefs all affected respondents’ agreement with related information on wildlife utilization. This research suggests that the publicity and education of scientific conservation methods should be emphasized in the future conservation education. In addition, scholars in the field of wildlife research should assume the role of ‘influencer’ and give full play to the scientific guidance of public opinion.

## 1. Introduction

The role of sustainable utilization of wildlife in promoting biodiversity conservation has always been a major focus of international debate [1]. Wildlife, as in the case of fisheries and forests, can be considered a renewable resource whose regenerative capacity allows for a degree of harvesting while maintaining populations at an ecologically viable level. A given harvest level is considered sustainable if it is at or below the level that would allow the resource to be regenerated permanently. The sustainable use paradigm promotes the managed use of living resources within sustainable limits to promote human wellbeing and basic needs [2,3]. However, activities such as trophy hunting and captive breeding which can contribute to conservation of biodiversity through financial funds (former) or by displacing consumer demand for products from wild sources (latter), [4,5] are increasingly in conflict with support for animal rights and welfare.

Among the conservation community there is a growing recognition that scientific and financial rationales for are no longer sufficient to protect and manage wildlife resources as values and opinions of the public increasingly diverge with traditional stakeholders such as conservation managers, and other land owners/users. Wider public support and participation is now considered essential to achieving the coordinated development of biodiversity conservation and sustainable utilization [6]. In some cases, such as trophy hunting and wildlife trade, attitudes and beliefs of stakeholders and the public may conflict with the plans proposed by wildlife managers or wildlife conservation scholars, resulting in management decisions being undermined by public opinion [7].

In recent years, China has attached great importance to advancing environmental conservation through the program known as ‘ecological civilization’, and wildlife management and conservation is attracting growing attention from all walks of life in Chinese society [8]. The widespread dissemination of information related to the wildlife utilization in new online media and traditional media will inevitably affect the public’s conservation concepts and attitudes, and thereafter trigger public discussion on the relationship between conservation and utilization [9]. Studies have shown that the same information can lead to different positive or negative reactions from the public depending on how it is presented and through which channels it is disseminated [10]. At present, the Chinese public’s awareness of wildlife conservation has been greatly enhanced, and some organizations have played an important role in this. Many of these organizations are committed to “compassionate conservation”, focusing on the welfare of individual animals rather than the sustainable development of wild animal species [11]. In addition, in the early days of the outbreak in Wuhan, China, some experts, based on incomplete experimental results, hastily concluded that bats, pangolins and other wild animals carried the virus and spread it to humans. They called on the public to ban all wild animals and blamed the COVID-19 outbreak in Wuhan on human consumption of wild animals [12].

It is imperative therefore that conservation managers and policy-makers are provided with more guidance and information about public attitudes and concepts with respect to wildlife conservation and management strategies, and to help them identify issues where there may be conflicts between public sentiment and rationale or traditional science based approaches that focus on species conservation and management. In the field of wildlife conservation and utilization, research on information as a topic itself focuses mainly on the impact of public opinion information on wildlife conservation and management decisions, and the impact of related information on consumer demand [13,14]. Public attitudes and behaviors towards biological conservation are related to their knowledge and understanding [15]. Therefore, conservation strategies, government activities, and other factors related to species described in media may affect online readers’ understanding of biological conservation and/or current management mechanisms [9]. With the progress of science and technology, mediums of information dissemination diversify. Both traditional media such as newspapers and radio, and new media such as web pages, search engines, and social software, have become effective mediums for information dissemination [16]. Mass communication as characterized by diversified subjects, timeliness of information production, immediacy of interaction, speed of dissemination, and massive content has increasingly become the engine to guide public opinion [17].

At present, research on the effect of information can be roughly divided into two categories, namely, (a) the analysis of factors that determine information dissemination in public media and (b) studies on what impact does the message have when it reaches the audience [18,19]. For example, Yinglin Wu (2018) collected and evaluated online news and public comments on China’s flagship species, the Chinese White Dolphin (*Sousa chinensis*), and analyzed the performance of media news in enhancing public awareness of conservation. Yinglin Wu (2018) analyzed how social media enhanced public awareness of conservation by collecting and evaluating online news and public comments on China’s flagship species, the Chinese white dolphin (*Sousa chinensis*). The results showed that most of the articles failed to popularize the knowledge of wildlife conservation, and even made the public highly suspicious of the conservation work of the government and experts [9]. However, as this study only focused on one species, and the content of relevant information and public responses may vary with species. We investigate and analyze the impact of widely disseminated and diverse content-oriented information related to the utilization of wildlife on the public’s concept of conservation and utilization of wildlife to explore the Chinese public’s attitudes towards sustainable utilization and management. Based on our research findings, we also provide some suggestions for wildlife conservation managers to optimize the information dissemination environment.

## 2. Materials and Methods

### 2.1. Data Collection

From 12 November to 4 December 2020, a stratified sampling method was used to investigate the impact of information related to wildlife utilization on the Chinese public using seven geographical regions in mainland China (Northeast, East, North, Central, South, Southwest, and Northwest). Each region has distinctive profiles in terms of their economic, cultural, social, ecological aspects of the development, as well as variety in terms of the depth and breadth of information radiation and penetration about issues pertaining to wildlife management and conservation [20].

A preliminary pilot test was conducted prior to the main study. The results of the pilot study were discussed with experts in the field of wildlife research, who reviewed and adjusted the questionnaire, which was then distributed online, using the platform “Questionnaire Star”. A total of 1645 questionnaires were collected. The questionnaires with ‘common sense trap questions’ were identified and excluded from the analysis as invalid. After screening, a total of 1294 valid questionnaires were obtained, with an effective rate of 78.7%. There were about 185 valid questionnaires for each of the seven geographic regions.

### 2.2. Questionnaire Design

Based on previous relevant research [21,22,23], the questionnaire consisted of two parts (see Appendix A). In the first part, there were 17 information topics related to wildlife utilization. These were identified from a comprehensive review of wildlife topics on major academic websites, online search engines, and social media platforms. Initially, 30 topics were identified and these were reduced for the survey to 17 following discussions with an expert group. These final 17 topics were divided into 4 categories: (1) information about preventing unsustainable and/or illegal utilization that clearly opposed excessive and illegal utilization of wildlife; (2) information against the any form of utilization of wildlife, including farmed wildlife species and species whose populations are not threatened; (3) information that supported sustainable utilization for certain purposes and based on science-based principles; (4) information promoting unsustainable utilization such as advertising that promotes demand for traditional wildlife products such as medical cures or therapy.

There were three questions under each topic. The first question enquired whether respondents had heard of the information topic. The second question enquired how respondents obtained the information about the topic, including Moments, WeChat groups, four major portals in China (Sina, NetEase, Sohu, and QQ), Microblog, TV, newspapers, etc. (If respondents had not heard of the first question, this question was skipped). The third one asked respondents’ attitude towards the information. We used the Likert 5-point scale method to assign values: strongly disagree = 1, basically disagree = 2, uncertain = 3, basically agree = 4, strongly agree = 5. In order to avoid any sequencing effects in the questionnaire, we randomly sorted the information topics on-line. The second part of the questionnaire investigated the demographics of respondents, including gender, age, place of residence, education level, vegetarianism, and religious beliefs.

### 2.3. Statistical Analysis

All data were processed and analyzed using SPSS 23.0 and Excel 2016. Cronbach’s Alpha was used to analyze the reliability of the survey results. KOM and Bartlett sphere tests were used to analyze the validity of the questionnaire. Cronbach’s Alpha was used for reliability analysis, concluding the reliability of 0.719. KOM and Bartlett sphere test were used to analyze the validity of the questionnaire items (KMO = 0.791 > 0.06, *p* = 0.000 < 0.05), indicating that the scale was suitable for testing by exploratory factor analysis. We ranked cities according to the 2020 City Business Charm Ranking released by China Business Network [24]. We used Excel to calculate demographic characteristics such as gender, age, place of residence, education level, occupation, monthly salary, vegetarianism, and religious beliefs, and calculated the average value of awareness and agreement with 17 items of information and 4 types of information in the questionnaire. The Chi-square test, one-way analysis of variance and independent sample *t* test were applied to analyze the differences in the awareness and agreement with the four types of information by different demographic variables. Based on this result, Pearson’s correlation coefficient was used to analyze the correlation between demographic variables and awareness and agreement. Finally, dummy variables were set for disordered categorical variables, and linear regression was used to test the influencing factors.

## 3. Results

### 3.1. Demographics

The basic information of respondents is shown in Figure 1. Male and female respondents numbered 575 and 719, respectively. Compared with the China Statistical Yearbook released by the National Bureau of Statistics in 2020, the ratio of men to women is relatively consistent [25]. Samples were relatively evenly distributed in first-tier cities, new first-tier cities, second-tier cities, third-tier cities, and other cities. The educational background of the respondents is higher than the average level in China [25]. Respondents with undergraduate education outnumbered those with primary school or junior high school education. The majority of respondents did not report being vegetarian or religious. Respondents under 45 years old accounted for 81.2%. We tried to evenly distribute the age of the respondents. However, as our questionnaire is published online and young people have more contact with the Internet, more young people participated in our survey. According to the 47th Statistical Report on China’s Internet Development released by the China Internet Network Information Center (CNNIC) in 2021, Internet users under the age of 50 accounted for 73.7% of China’s total Internet users, hence from this perspective, our sample is representative [26].

### 3.2. Respondents’ Awareness of Related Information on Wildlife Utilization

Respondents’ average awareness of 17 topics, grouped under four categories is shown in Table 1. The respondents’ awareness of information on preventing unsustainable and illegal utilization was the greatest, reaching 87.6%; awareness of information that against utilization was 67.5%; awareness of information that supported sustainable utilization was 60.7%; the awareness of information about promoting unsustainable utilization was the lowest, only 17.1%. Respondents had the greatest awareness of information on preventing unsustainable and illegal utilization and the least awareness of information on promoting unsustainable utilization. It is noteworthy that the public had more awareness of information against utilization than information supporting sustainable utilization.

The top six information subjects with the greatest awareness included were “Public service message: When the buying stops, the killing can too”, “Pangolin is one of the world’s most trafficked mammals and is on the verge of extinction due to human’s consumption”, “Consumption of wildlife and wildlife-turned products, most of which do not go through quarantine and other preventive procedures, may cause the highly pathogenic zoonosis”, “The COVID-19 epidemic sounded the alarm that consumption of wildlife should be banned”, “Animals are widely utilized in experiments in fields such as life science and medicine, making a great contribution to the scientific progress and healthy life of humanity”, and “The overfishing of sharks due to Chinese people’s demand for fins is threatening the survival of the species”. The primary channels for the receiving information among the top six are shown in Figure 2. Television was the most influential channel for the dissemination of information related to wildlife utilization, followed by WeChat moment, QQ group, WeChat group, WeChat official accounts, four major portals and Microblog, China’s microblogging platform.

The survey instrument included three items of information about COVID-19 epidemic and wildlife. The respondents’ average awareness of such information was 73.65%, indicating that most of them had read about or heard about it. Specifically, the public had the greatest awareness of “Consumption of wildlife and wildlife-turned products, most of which do not go through quarantine and other preventive procedures, may cause highly pathogenic zoonosis” (86.4%); “COVID-19 epidemic sounded the alarm that consumption of wildlife should be banned” (81.2%); and “Given the health of 1.4 billion people, the interests of those who raise wildlife for human consumption is insignificant” was less known than the former two items of information, which exceeded 50% (53.3%).

### 3.3. Respondents’ Agreement with Information Related to Wildlife Utilization and Influencing Factors

Respondents’ average agreement with the17 individual topics of information under four categories is shown in Table 2 based on the following Likert scale: strongly disagree = 1; disagree = 2; uncertain = 3; agree = 4; and strongly agree = 5. The higher the average result, the higher agreement. Respondents showed the greatest agreement for information on preventing unsustainable utilization (4.5) and the lowest agreement for information on promoting unsustainable utilization (2.6); furthermore, their agreement with information that against utilization (3.9) was higher than that for information which supported sustainable utilization (3.7).

Regression analysis was performed by using difference analysis and correlation analysis and males were set as a reference variable for linear regression. The results showed that females had a higher agreement with anti-utilization information than males (β = 0.110, *p* = 0.000), while their agreement regarding sustainable utilization was lower than males (β = −0.099, *p* = 0.004). Females had a higher agreement about preventing unsustainable utilization information than males (β = 0.063, *p* = 0.041). With vegetarianism as the reference variable, the conclusion was that vegetarians’ agreement with preventing unsustainable utilization information was higher than non-vegetarians (β = −0.148, *p* = 0.000). With religious beliefs as the reference variable, it was found that religious followers’ agreement with preventing unsustainable utilization information was higher than non-followers (β = −0.100, *p* = 0.019). The younger respondents were, the higher the agreement with information that against utilization (β = −0.048, *p* = 0.000) and the higher the agreement with preventing unsustainable utilization information (β = −0.060, *p* = 0.000). The more developed a city was, the higher the agreement with preventing unsustainable utilization information (β = −0.040, *p* = 0.004). Respondents with higher education level had higher agreement with preventing unsustainable utilization information (β = 0.052, *p* = 0.000) and lower agreement with information about promoting unsustainable utilization (β = −0.080, *p* = 0.002).

The top six pieces of information or phrases/slogans with the greatest agreement included: “In most cases, wildlife and related products from illegal sources, have not gone through quarantine and other preventive procedures. Consumption of them may cause the highly pathogenic zoonosis”, “Public service message: When the buying stops, the killing can too”, “Pangolin is one of the world’s most trafficked mammals and is on the verge of extinction due to human’s consumption”, “Animals are widely utilized in experiments in fields such as life science and medicine, making a great contribution to the scientific progress and healthy life of humanity”, “Hunting, for whatever purpose, is cruel and inhuman since it violates animals’ right to survival”, and “Raising bears to extract their bile is maltreatment”.

Respondents’ average degree of agreement with information related to COVID-19 and wildlife utilization was 4.23. Moreover, 75.73% of the respondents agreed with the phrase “Given the health of 1.4 billion people, the interests of those who raise wildlife for human consumption is insignificant”, while 536 of them strongly agreed; only 126 respondents basically disagreed or strongly disagreed with this item. A total of 94.05% of the respondents agreed the phrase “Consumption of wildlife and related products from illegal sources may cause the highly pathogenic zoonosis”. Furthermore, 77.67% of the respondents agreed that “The COVID-19 epidemic sounded the alarm that consumption of wildlife should be banned”, and 570 of them expressed strong agreement; only 132 respondents basically disagreed or strongly disagreed (Figure 3).

## 4. Discussion

According to the data, respondents had the greatest awareness and agreement with information on preventing unsustainable utilization and illegal utilization and the least awareness and agreement with information on promoting unsustainable utilization. Moreover, people living in more developed cities have a higher agreement with information on preventing unsustainable utilization, and the more educated the people are, the more they agree with information on preventing unsustainable utilization and the less they agree with information on promoting unsustainable utilization. Therefore, the public is clearly opposed to wildlife smuggling, hunting and other illegal utilization activities. However, they also can carefully judge more extreme information from advertisements and rumors and myths about the exaggerated efficacy of wildlife medicines and health products. This is a remarkable achievement of China’s promotion of science education on nature and environment under the ‘ecological civilization’ program. However, it is also noteworthy that respondents’ have higher levels of awareness and agreement with information that does not support utilization than the case with information about support for sustainable utilization and we explore this further below.

### 4.1. Respondents’ Awareness of Information That Do Not Support Utilization Is Higher That of Information Which Support Sustainable Utilization

We found that respondents’ awareness of information that does not support utilization is higher than that of information which support sustainable utilization. However, there are also many voices supporting sustainable utilization. Of the nine information topics that did not support utilization selected, seven arise from campaigns by domestic and international NGOs, and these topics have the greatest average awareness (Table 1). In recent years, some animal conservation organizations been very active popularizing their messages on a variety of media, acting as direct sources of information and also providing press releases that are adopted verbatim needed for news production through formal or informal channels [27]. Many animal conservation organizations therefore can also call on the power of the media to rally public opinion and mobilize social resources to their chosen causes. Taking advantage of the influence of celebrities, well-known entrepreneurs, senior media experts and other third parties with a high profile, ensures their views are widely disseminated and adopted. Some of these animal conservation organizations also secure the support of the public by clandestine approaches such as pretending to be members of the general public to further influence public opinion, thus affecting government decisions on wildlife conservation and utilization management. In effect they have become de facto influencers, main sources of information and policy advisors on all wildlife-related information [28,29,30].

The role of celebrities from TV and sport is incredibly important and in China, they have more freedom to speak out in social media due to their profile. With a large number of followers, they can easily become the representatives of NGOs for wildlife-related information as information disseminated through them will be amplified [31,32]. For example, CCTV and several local TV stations frequently broadcast slogan against the consumption of tiger products, ivory, shark fins and other products with the title “When the buying stops, the killing can too” by international superstars such as Leonardo DiCaprio, Jackie Chan, and Yao Ming in 2011, which brought about a profound impact. In this study, respondents have a high degree of awareness of this particular slogan: “When the buying stops, the killing can too”; with 94.5% stating that they have heard of or seen this advertisement, mainly from television. The related information “The overfishing of sharks due to Chinese people’s demand for fins is threatening the survival of the species” is known of 73.4% of respondents. Our r results certainly suggest that influencers, represented by celebrities, appear to be exerting a significant impact on the public’s awareness of information related to wildlife utilization in China.

The influencer is an important concept in communication science, and refers to activists who often provide information for others in interpersonal communication networks and influence the behavior and attitudes of others. They play an important mediating role in the formation of mass communication effects as important drivers of the formation of many public opinions [32,33,34]. Such influencers, at the center of public opinion, selectively access information and knowingly or unknowingly interpret it through their filtering role and integrate and process information for active communication, to deliberately influence the thinking of their followers. A study by Choi (2009), for example, showed that compared to non-influencers, influencers showed higher motivation to join communities with higher levels of community participation and more frequent network relationship formation [35]. Influencers concerned with wildlife conservation mainly include animal conservation NGOs, science popularizers, wildlife conservation volunteers, celebrities, well-known entrepreneurs and senior media professionals [36]. Influencers concerned with wildlife conservation can enhance public awareness of wildlife conservation and stimulate public enthusiasm for wildlife conservation, but some of them lack knowledge of the complex scientific nature of conservation issues as often they are professionals from other fields who lack scientific knowledge of wildlife conservation [37]. They sometimes engage with and pass judgement on contentious issues outside their professional experience with enthusiasm, but without confirming its authenticity or scientific validity, often with strong emotional overtones. Under this circumstance, they can amplify a very one-sided interpretation of conservation issues which favors emotional incitement of their followers over drier, science based information and rationality [38].

The influence of new online media is generally higher than that of other traditional media such as newspapers and radio, except for television. According to the 47rd Statistical Report on the Development Status of the Internet in China released by the China Network Information Center (CNNIC), as of December 2020, China’s Internet users has increased drastically and now numbers in excess of 989 million [26]. New media has flourished due to its advantages of speed, timeliness and convenience of interaction, eclipsing the influence of traditional media [39]. The growth in social media and the recent upsurge in self-media, the threshold for detailed science-based content in output has been significantly curtailed the opportunity for more rationale and nuanced discussion greatly diminished. Professional, mainstream media, once the most important source of information on conservation has been strongly impacted by the non-professional self-media in terms of quantity, timeliness and influence [40]. In the self-media environment, influencers’ subjective expressions of opinion further keep the public silent and accepting of the influence of influencers [41]. Some studies have shown that these convenient social platforms have a significant impact on public awareness of wildlife conservation [42], and that raising public awareness through social media can enhance wildlife conservation and management [7]. Therefore, further research on the role of new media in wildlife conservation management is necessary in the future.

### 4.2. Respondents’ Agreement with Information That Does Not Support Utilization Is Higher That of Information Which Supports Sustainable Utilization

According to the results, respondents’ agreement with information that do not support wildlife utilization is higher than that of information which support sustainable utilization. The German scholar Noelle Neumann, says that when people feel that their opinions belong to the “majority”, they are more inclined to express them actively and boldly; when they find that their opinions belong to the “minority”, they may remain silent for fear of being isolated. If this cycle continues, one side will become more powerful while the other tends to keep silent [43,44]. Given the increased dominance of social media amongst younger generations and deep demographic changes and depopulation of rural areas, there are fewer and fewer people with knowledge and experience of wildlife management the large scale migration there seems to be an inevitability that voices supporting sustainable utilization and management will fade due to disparities in numbers, wealth, power, and attention, and conservation managers will face an increasingly challenging policy environment to maintain sustainable management [45].

The argument against any form of utilization overlooks the critical role of regulated utilization in protecting wildlife populations in the wild. ‘Compassionate Conservation’ as advocated by animal rights groups is very likely to trigger emotional agreement [42,46], which tends to guide the public to adopt a firm attitude that opposes wildlife utilization in all forms and purposes. However, there is still much public support for the utilization of certain wildlife species, and its products from captive breeding sources; some of which is influenced by traditional cultural backgrounds and popular science education in China. For example, one study found that the basic attitude of the Chinese public towards the conservation of snakes was in favor of their moderate utilization in medicine but objecting to their consumption as food [47]. Our study reveals similar results. This study shows that the public is highly supportive of the widespread utilization of animals in experiments in fields such as life sciences and medicine, despite being influenced by organizations claiming that “animal experimentation is unethical and immoral”. Similarly, the public, while susceptible to the ‘news’ of animals being abused for performance on social media, is still supportive of zoos and wildlife parks for nature conservation education, wildlife breeding stock preservation, translocation and scientific research.

Communication research suggests that the more credible a source is perceived to be, the more likely the audience is to be persuaded. Usually, sources from authoritative organizations and experts have high credibility [48,49]. The COVID-19 epidemic has been rampant worldwide, causing great harm. Some experts say that this epidemic has once again sounded a stark alarm about the illegal consumption and utilization of wildlife [50]. Therefore, the state should adopt a “one size fits all” ban on the wildlife farming for food considering the safety of 1.4 billion people and the fundamental interests of the country [51]. This research shows that majority of the public are aware that the consumption of illegally sourced wildlife and its products is unsafe and can cause highly pathogenic anthropozoonosis. Moreover, there is generally a high level of public agreement with information delivered by experts, with majority of the public agreeing that all wildlife should be banned in order to ensure public health safety. However, the study by Booth in 2021 demonstrates that the abrupt removal of wildlife meat from the food system may have negative effects on human and nature, and we need to consider the remote coupling between food systems and nature in formulating wildlife trade policy interventions [52]. Virus traceability is a serious issue to be studied on a rigorous and factual basis by scientists and medical experts who uphold an evidence-based line of inquiry and a spirit of cooperation [53]. The appropriateness of imposing a “one size fits all” ban on wildlife farming in the absence of a definitive conclusion remains to be explored, but our research shows that the public has undoubtedly stood on the opposite side of those who raise wildlife for food.

There is a high level of public agreement with the slogan “When the buying stops, the killing can too”. A number of celebrities have promoted this concept in China working with international animal conservation organizations, and there have been loud calls on the Chinese public to stop consuming wildlife and their products, even including species which are not threatened or derived from captive breeding sources. For example, a survey by WildAid found that their media campaign with celebrities calling for an end to the consumption of shark fins had a significant impact, with 83% of those who saw the campaign messages having stopped or reduced their consumption [54]. Public agreement with the information that “Raising bears to extract their bile is maltreatment.” is also extremely high, with only 106 out of 1294 respondents disagreeing or strongly disagreeing. Other studies are similar, one finding that most people believe that artificial bile extraction is harmful or very harmful to the health of bears, and this is likely to be a major reason for people rejecting bear bile products [55]. After 2011, the campaign featuring saving black bears, banning bile products, and opposing bear farming for live bile led by animal conservation organizations, has been met with an overwhelming response on platforms such as Microblog, with strong support from major media outlets [56].

Our research also found that demographic factors affect the public’s agreement with information related to wildlife utilization. Previous studies have shown that gender influences public perceptions of wildlife conservation, with men being more supportive of sustainable wildlife conservation management [57,58]. In this study, we also found that women agree more with information that does not support utilization, while men are more aware that regulated utilization contributes to sustainable wildlife development, and agree more with information that support sustainable utilization. Similarly, age is also an important factor. Kellert and Berry suggest in their 1981 study that younger people are more opposed to animal use than older people, and older people tend to emphasize the practical value of animals [59]. Our research also shows that young people are more likely to get information through social media, and the younger and less experienced they are, the more likely they are to be influenced by information against utilization [60]. Not surprisingly, vegetarians and people with religious beliefs agree more with information that does not support utilization than non-vegetarians and people without religious beliefs, and other studies have confirmed this [61].

### 4.3. Provide More Balanced Information about Sustainable Approaches to Conservation through Use of ‘Influencers’ Operating on a Scientific Basis

This research also shows that influencers such as animal conservation NGOs, experts, celebrities, science workers, and volunteers with a large following, have a central role to play and can enhance public awareness of wildlife conservation issues and inspire others to become involved. While attention to the environment can increase public awareness of the environment, unbalanced information may instead lead to misunderstandings about the decisions or actions of governments and experts, and may undermine public support for science-based environmental policies advocated by governments [7]. In the era of “we-media” and “all-media”, it is important to develop new synergies of information and knowledge that acknowledge and promote new understandings about the role of animal welfare and rights in contemporary science led conservation practice and policy [62]. A key factor will be to develop online messaging that is appealing, visually appealing and easily understood by social media users.

It is imperative therefore, that experts and scholars in the field of wildlife conservation should invest more time and resources in establishing themselves as influencers to provide a guiding and mentoring role in a wider variety of media channels, especially influential social media platforms, such as using Microblog or WeChat Public Platform [63]. In this way a more balanced and nuanced understanding about controversial issues such as artificial breeding and utilization and its impact on wild populations can be sought, Easy-to-understand content to reflect contemporary scientific understandings of wildlife conservation and management is essential to advance the complex relationship between wildlife conservation and utilization [31].

## 5. Conclusions

This study is the first to explore the impact of information of wildlife utilization on public conservation concepts in China by presenting information content across a range of issues directly in questionnaires. By adopting the stratified sampling method, this study has improved the representativeness of the sample as much as possible, so that the characteristic structure of the sample’s respondents is basically consistent with the overall population. On the whole, the Chinese public had the greatest awareness and agreement with information on preventing unsustainable utilization and illegal utilization and the least awareness and agreement with information on promoting unsustainable utilization. It is noteworthy that Chinese public have higher levels of awareness and agreement with information that does not support utilization than the case with information about support for sustainable utilization. From our research, we can see that under the popular appeal of information producers and influencers in the information generation and transmission process the majority view in China is closely aligned with a model of wildlife conservation that does not necessarily include wildlife utilization. This finding is especially significant among young people. Therefore, we believe that in the future conservation education must find ways to provide a deeper and more nuanced discourse regarding conservation management and policy by encouraging conservation scientists to make fuller use of social media platforms and tools to communicate their findings in an accessible and attractive way.

## Figures and Tables

**Figure 1 animals-11-02640-f001:**
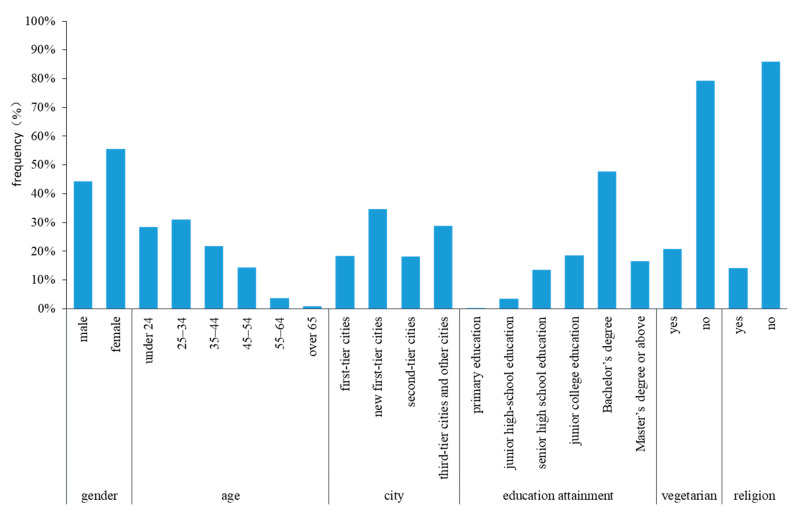
Demographic bar chart.

**Figure 2 animals-11-02640-f002:**
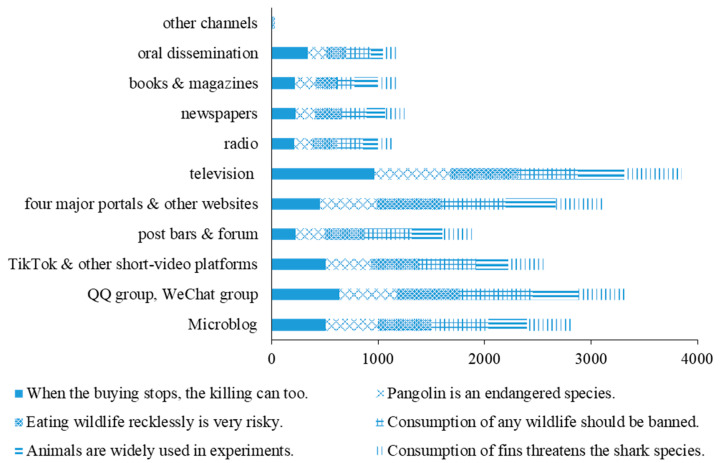
The channel through which respondents obtain information about the top 6 topics.

**Figure 3 animals-11-02640-f003:**
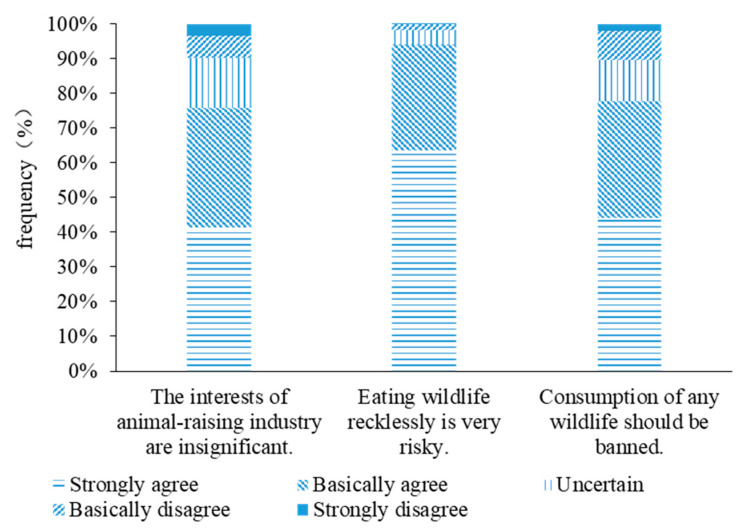
Public agreement to information related to COVID-19.

**Table 1 animals-11-02640-t001:** Average information awareness.

Information Types	Information	Information Awareness (%)	Average Information Awareness (%)
Preventing unsustainable/illegal utilization	Pangolin is one of the world’s most trafficked mammals and is on the verge of extinction due to human’s consumption.	88.79	87.60
	In most cases, wildlife and related products from illegal sources, have not gone through quarantine and other preventive procedures. Consumption of them may cause the highly pathogenic zoonosis.	86.40	
Against utilization	Public Service Message “When the buying stops, the killing can too.”	94.51	67.51
	Given the health of 1.4 billion people, the interests of those who raise wildlife for human consumption is insignificant.	53.32	
	Animal experimentation should be forbidden since it is unethical and immoral.	53.94	
	The overfishing of sharks due to Chinese people’s demand for fins is threatening the survival of the species.	73.42	
	Raising bears to extract their bile is maltreatment.	68.24	
	Hunting, for whatever purpose, is cruel and inhuman since it violates animals’ right to survival.	66.46	
	Every link such as breeding, transportation and selling in the chain of exotic pet trade inflicts great pain on animals.	53.63	
	Captivity of animals in zoos goes against their nature and the cruel training always comes with abusing.	62.83	
	“The COVID-19 epidemic sounded the alarm that consumption of wildlife should be banned.”	81.22	
Support sustainable utilization	Animals are widely utilized in experiments in fields such as life science and medicine, making a great contribution to the scientific progress and healthy life of humanity.	76.28	60.68
	The industry of raising fur-bearing animals (such as mink and fox) plays an active role in wildlife conservation.	54.33	
	As an important part of traditional Chinese medicine, medicines obtained from animals can contribute to the healthcare of humanity.	65.61	
	Zoos and safari parks can serve to educate the public on nature conservation, promote ecological culture, provide public entertainment, conserve wildlife germplasm, offer off-site conservation and scientific research.	67.93	
	Orderly hunting under management is an effective way of conservation.	39.26	
Promoting unsustainable utilization	Mobula gills are anti-carcinogenic.	17.16	17.16

**Table 2 animals-11-02640-t002:** Average Value of Agreement with Information.

Information Types	Brief Information	Information Agreement	Average Information Agreement
Preventing unsustainable/illegal utilization	Pangolin is an endangered species.	4.440	4.500
	Eating wildlife recklessly is very risky.	4.560	
Against utilization	When the buying stops, the killing can too.	4.480	3.943
	The interests of animal-raising industry are insignificant.	4.042	
	We should oppose animal experimentation.	3.085	
	Consumption of fins threatens the shark species.	3.996	
	Raising bear for its gallbladder and bile is maltreatment.	4.116	
	Hunting is inhuman behavior.	4.121	
	Pet industry chains harm animals.	3.843	
	Zoos abuse animals.	3.709	
	Consumption of any wildlife should be banned.	4.097	
Support sustainable utilization	Animals are widely used in experiments.	4.210	3.738
	Raising fur-bearing animals helps to protect wildlife.	3.717	
	Medicines obtained from animals are important.	3.589	
	Zoos play important roles.	4.087	
	Orderly hunting under management is an effective way of conservation.	3.087	
Promoting unsustainable utilization	Mobula gills are anti-carcinogenic.	2.589	2.589

## Data Availability

The data presented in this study are available on request from the corresponding author. The data are not publicly available due to privacy and ethical restrictions.

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
