# Peer review of "The Impact of Information on Attitudes toward Sustainable Wildlife Utilization and Management: A Survey of the Chinese Public"

_animals, 2021, doi:10.3390/ani11092640_

Round 1
Reviewer 1 Report
22 – I would use “greatest” rather than “highest” (throughout)
I would suggest that the simple summary is not really all that simple. Can the authors develop a summary that is easy to interpret for those with limited training? I have a fairly good idea of what the paper will be about, but I am left with many questions after reading this summary.
35 – better word than purify?
50-51 – this sentence is unclear: “…we should emphasize the interpretation of the inherent tension between conservation and utilization in future wildlife conservation education.” I’m not sure why the interpretation, specifically, should be emphasized.
Abstract should include the number of questionnaires distributed and the response rate.
73-78 – These two sentences seem to be a key justification for your study. I would suggest that you elaborate or clarify the two “positions” (conservation vs. utilization).
97 – the term “survival” seems out of place
109-110 – I would say that this statement is false. It may be true, however, if the statement is limited to social media.
127 – What was the stratification factor? This is important relative to the results shown in Fig. 1.
Methods – You should include the total number of surveys that were distributed.
136 – A question about Questionnaire Star. No knowing the platform, are there any biases regarding who could respond to the survey? Given my American bias about what I think I know about the ways in which the Chinese people interact with wildlife, I am surprised by some of the findings, particularly the relatively high level of agreement “against utilization.” Can the authors provide some assurance that their sample is representative of the larger Chinese population?
139 – Is effective rate the same as the effective response rate?
155 – I wonder if “information topic” is the correct way to categorize the survey questions. They feel more like statements, or belief statements.
Table 1 – splitting the responses into the four categories doesn’t seem like a legitimate way to categorize responses when your greatest and least values only had 2 and 1 questions, respectively.
Table 1 – the statement Public Service Message “When the buying stops, the killing can too.” Seems like it is in the wrong category given what is written on lines 201-203.
Fig. 2 – Are there any channel-information interactions?
289 – should “that” be “than”?
293-295 – This is a confusing statement that isn’t really highlighted in the results. Given that this is a major result of the paper, can it be stated more clearly in the results section?
368-370 – Same comment as above.
436-437 – This is the first time that the concept of an “emotional” reaction has been brought up. I’m not sure it belongs here given that there are no questions that actually address this, only whether there is agreement with a particular statement.
438-439 – Similar argument to above but with men and rationality. I don’t think this is supported by the data.
459-462 – This is a sexist statement. I realize that the authors are trying to “prove” that there is value in sustainable harvest, but there is also value in animal rights and ethical animal treatment. If this statement remains in the paper, it should be followed by one that states the ways in which men need to be educated to have greater compassion for animals.
480-483 – I will admit that I write this review from an American perspective, but I think that it is extremely important to understand that the values and beliefs (and emotions) of the public are equally as relevant as the science when we talk about conservation and management decisions. Scientists need to be better communicators, rather than thinking that the public simply needs to be educated to ensure that correct management decisions are made.
Author Response
General:
It should be noted that the authors undertook quite interesting research of a sociological nature, the results of which should have both application and practical significance in terms of forms and activities of natural resource management and protection. Proper shaping of public opinion plays in many cases a key role in the planning and implementation of multi-directional activities related to the protection of natural resources.
However, after reading the manuscript, I came up with a few comments that should improve the study. I propose to make minor corrections in line with my suggestions and accept the manuscript for publication.
Reply: Thanks for the reviewer's affirmation of the paper and the suggestions for this paper. According to your suggestion, we mainly revised the Discussion and Conclusions of the article.
Specific:
Comment 1: Line 55 - in my opinion, the key fame should be supplemented with: wild animals.
Reply 1: Thanks for your suggestion. We added the key word: wild animals. (Ln 57)
Comment 2: Figure 1 - in my opinion, the participation of young people in the study is too high. People up to 34 years of age account for over 60% of the entire sample. This is important because people who, as a rule, do not have any influence on management activities in the field of animal protection and sourcing. Authors should articulate this in the manuscript.
Reply 2: Yes, that's an important point you mentioned. We have tried our best to evenly distribute the age of the respondents. However, as our questionnaire is published online and young people have more contact with the Internet, more young people participate in our survey. According to the 47th Statistical Report on Internet Development in China released by CNNIC in 2021, Internet users under the age of 50 account for 73.7% of China's total Internet users. We have added relevant explanations in the article. (Ln 205-211)
Comment 3: Line 211-212 - trivialism, is it a well-known element?
Reply 3: Thanks for your advice, we deleted that sentence.
Comment 4: Line 220-221 - I don't really see the connection, but if so, the respondents gave the answer?
Reply 4: In the early days of the COVID-19 outbreak in Wuhan, China, some experts believed that wild animals, such as bats and pangolins, carried the virus and passed it on to humans. They have called on the public to ban all wild animals, blaming human consumption of wild animals for the COVID-19 outbreak. "The COVID-19 epidemic sounded the alarm that consumption of wildlife should be banned" is the quote of an expert in an interview. The expert's comments have had a strong impact on the Chinese public, and results show that respondents have a high degree of agreement with this sentence. We made this clear in the introduction (Ln 93-98).
Comment 5: Line 268-269 - very strange again, but does that mean people don't understand the essence of the COVID 19 problem?
Reply 5: Yes, in the early days of the COVID-19 outbreak, the tracing of the source was not finished, and the public did not understand its nature.
Comment 6: Line 276-290 - this is not a discussion, but conclusions ????
Reply 6: The main results of our study are as follows: 1) The respondents have the greatest awareness and agreement of the information about preventing unsustainable utilization and illegal utilization; 2) Participants had the lowest awareness and agreement of the information about promote unsustainable utilization; 3) The awareness of the information that does not support the utilization is greater than the awareness of the information that supports the sustainable utilization; 4) The agreement of respondents with the information that against utilization is greater than the agreement with the information that supports sustainable utilization. We believe that the first two results are not the focus of this study, so we only discuss briefly in this paragraph, and then propose to focus on the last two results in the following paragraphs. We have modified the conclusion according to your suggestion. (Ln 500-518)
Comment 7: Line 303 - Taking advantage of the influence of celebrities - is it a problem in shaping public opinion in many countries and distorting reality?
Reply 7: The influence of celebrities is a powerful driving force in the dissemination of information. The more influential celebrities are involved in the dissemination of information, the better the dissemination effect will be. And that influence can be used by people with ulterior motives.
Comment 8: Line 323-325 - this is the case in many countries of the world, which should be treated as a kind of hypocrisy, because it is these celebrities who do not respect the principles of nature protection to the greatest extent (!)
Reply 8: Yes, these celebrities often do not have the professional knowledge of the scientific protection of wild animals, they do not know what the principles of nature conservation is.
Comment 9: Line 432 - it needs to be strongly emphasized, but does this result from the content of the research?
Reply 9: This conclusion is not the content of this study, we have added references, thank you very much for your advice. ([56] Zhao, Z.; Wei, M. Research on ngos' intervention strategies in public opinions:an analysis based on guizhentang event. J. Nanjing U. Posts. Te. 2012, 12, 1-6.).
Comment 10: Line 374-380 - very correct statements, despite the fact that they do not result directly from the research carried out.
Reply 10: Thank you for the positive feedback.
Comment 11: Line 408 - Booth (2021) - Formatting?
Reply 11: Thanks for your suggestion, we changed the format. (Ln 435)
Comment 12: Line 441-443 - confirmation of my previous comment.
Reply 12: Yes, we have added the corresponding instructions in the manuscript. (Ln 467-472)
Comment 13: The chapter Strengthen the publicity and education of the relationship between conservation and utilization, and give play to the positive guiding role of influencers - is not a discussion, only statements, or even conclusions, but not entirely derived from the content of the manuscript, it necessarily requires improvement.
Reply 13: Yes, we agree with you. We have modified this part. (Ln 476-498)
Comment 14: Line 476 - the conclusions should be corrected because, although I agree with them, they do not result directly from the research carried out.
Reply 14: Thanks for your advice. We rewrote the conclusion (Ln 500-518).

Reviewer 2 Report
It should be noted that the authors undertook quite interesting research of a sociological nature, the results of which should have both application and practical significance in terms of forms and activities of natural resource management and protection. Proper shaping of public opinion plays in many cases a key role in the planning and implementation of multi-directional activities related to the protection of natural resources.
However, after reading the manuscript, I came up with a few comments that should improve the study:
Line 55 - in my opinion, the key fame should be supplemented with: wild animals,
Figure 1 - in my opinion, the participation of young people in the study is too high. People up to 34 years of age account for over 60% of the entire sample. This is important because people who, as a rule, do not have any influence on management activities in the field of animal protection and sourcing. Authors should articulate this in the manuscript.
Line 211-212 - trivialism, is it a well-known element?
Line 220-221 - I don't really see the connection, but if so, the respondents gave the answer?
Line 268-269 - very strange again, but does that mean people don't understand the essence of the COVID 19 problem?
Line 276-290 - this is not a discussion, but conclusions ????
Line 303 - Taking advantage of the influence of celebrities - is it a problem in shaping public opinion in many countries and distorting reality?
Line 323-325 - this is the case in many countries of the world, which should be treated as a kind of hypocrisy, because it is these celebrities who do not respect the principles of nature protection to the greatest extent (!)
Line 432 - it needs to be strongly emphasized, but does this result from the content of the research?
Line 374-380 - very correct statements, despite the fact that they do not result directly from the research carried out.
Line 408 - Booth (2021) - Formatting?
Line 441-443 - confirmation of my previous comment.
The chapter Strengthen the publicity and education of the relationship between conservation and utilization, and give play to the positive guiding role of influencers - is not a discussion, only statements, or even conclusions, but not entirely derived from the content of the manuscript, it necessarily requires improvement.
Line 476 - the conclusions should be corrected because, although I agree with them, they do not result directly from the research carried out.
I propose to make minor corrections in line with my suggestions and accept the manuscript for publication.
Author Response
Comment 1: 22 – I would use “greatest” rather than “highest” (throughout)
Reply 1: We appreciate this comment and the reviewer’s concerns. We changed “highest” to “greatest” throughout.
Comment 2: I would suggest that the simple summary is not really all that simple. Can the authors develop a summary that is easy to interpret for those with limited training? I have a fairly good idea of what the paper will be about, but I am left with many questions after reading this summary.
Reply 2: Agreed. We have noted this, and also modified the Summary. (Ln 16-31)
Comment 3: 35 – better word than purify?
Reply 3: We modified “purify” to “optimize”. (Ln 37)
Comment 4: 50-51 – this sentence is unclear: “…we should emphasize the interpretation of the inherent tension between conservation and utilization in future wildlife conservation education.” I’m not sure why the interpretation, specifically, should be emphasized.
Reply 4: Thanks for your suggestion. From our research, we can see that the current Chinese public's concept of wildlife protection tends to be absolute protection, and this trend is especially significant among young people. Therefore, we believe that in the future conservation education, we should not only continuously improve the public's respect and love for nature, but also pay attention to the publicity and education of scientific conservation methods. We modified the sentence. (Ln 53-54)
Comment 5: Abstract should include the number of questionnaires distributed and the response rate.
Reply 5: Thanks for your comment. Our questionnaires are distributed online, and we have added the total number and efficiency of the questionnaires in the abstract (Ln 41-42).
Comment 6: 73-78 – These two sentences seem to be a key justification for your study. I would suggest that you elaborate or clarify the two “positions” (conservation vs. utilization).
Reply 6: We fully agree more detail on this topic was needed. I elaborated on both sides of the argument. (Ln 68-71)
Comment 7: 97 – the term “survival” seems out of place
Reply 7: Agreed, we deleted the word.
Comment 8: 109-110 – I would say that this statement is false. It may be true, however, if the statement is limited to social media.
Reply 8: Thank you for your spotting this discrepancy, and we deleted the sentence.
Comment 9: 127 – What was the stratification factor? This is important relative to the results shown in Fig. 1.
Reply 9: The survey was stratified according to the seven geographical regions of mainland China, and an equal number of questionnaires were distributed in each geographical region. We identified the stratification factor in the manuscript (Ln136-139).
Comment 10: Methods – You should include the total number of surveys that were distributed.
Reply 10: Our questionnaire was sent online through Questionnaire Star, and we could only see how many people had completed and submitted the questionnaire. The system showed that a total of 1645 people completed the questionnaire. Incorrect answers to randomly inserted common sense trap questions were invalid questionnaires, and 1294 valid questionnaires were obtained after screening.
Comment 11: 136 – A question about Questionnaire Star. No knowing the platform, are there any biases regarding who could respond to the survey? Given my American bias about what I think I know about the ways in which the Chinese people interact with wildlife, I am surprised by some of the findings, particularly the relatively high level of agreement “against utilization.” Can the authors provide some assurance that their sample is representative of the larger Chinese population?
Reply 11: Questionnaire star has no bias about who can respond to the survey. The questionnaire star randomly invited some people who had participated in questionnaire filling to join the sample library. The sample source was random and the sample was true. Questionnaire star provides customized sample service for customers. Surveyors can request the minimum number of questionnaires to be collected and the minimum number of participants of each type. Questionnaire star can issue questionnaires to those who meet the requirements according to our requirements for the target population. This study adopts the method of stratified sampling to improve the representativeness of the sample as much as possible, so that the characteristic structure of the sample respondents is basically consistent with the overall structure. In result 3.1, we supplemented the comparison between the characteristic structure of the respondents and the whole. (Ln 199-212)
Comment 12: 139 – Is effective rate the same as the effective response rate??
Reply 12: The "efficiency rate" here refers to the percentage of all returned questionnaires that answered correctly the randomly inserted common sense trap question. See Reply 10.
Comment 13: 155 – I wonder if “information topic” is the correct way to categorize the survey questions. They feel more like statements, or belief statements.
Reply 13: The classification of information was divided into four categories according to the level of support for the wildlife utilization contained in the information. Do not support the wildlife utilization was divided into not support unsustainable use and do not support all use; Support for use was divided into support for sustainable use and promotion of unsustainable use. The third category is "Support sustainable utilization" instead of "Support unsustainable utilization" which we miswrote and have been corrected. We have explained each type of information. (Ln 158-165)
Comment 14: Table 1 – splitting the responses into the four categories doesn’t seem like a legitimate way to categorize responses when your greatest and least values only had 2 and 1 questions, respectively.
Reply 14: Thanks for your comment. After a lot of collection and discussion, we finally determined the four types of information, a total of 17. In the process of collecting, we found that under the policy constraints and the supervision of public opinion, there is little information to promote unsustainable utilization. Among the messages against the wildlife utilization, there are also very few messages explicitly against unsustainable use, and most of the messages express the meaning of absolutely not supporting the wildlife utilization, or even resisting it. Therefore, the number of these two types of information in the final filter is small.
Comment 15: Table 1 – the statement Public Service Message “When the buying stops, the killing can too.” Seems like it is in the wrong category given what is written on lines 201-203.
Reply 15: Thank you for reminding. Although the slogan "When the buying stops, the killing can too." focuses on the threatened species of the population, literally, the aim of this slogan is "the buying stops ", that is, it absolutely does not support wildlife utilization. Therefore, it belongs to the second category. We have modified the paragraph. (Ln 226-235).
Comment 16: Fig. 2 – Are there any channel-information interactions?
Reply 16: These channels must be interacting. So, in our questionnaire, " Through what channel did you learn about this information?" The question is multiple choice.
Comment 17: 289 – should “that” be “than”?
Reply 17: Yes, thank you very much for your attention and reminding. We have made modifications. (Ln 316)
Comment 18: 293-295 – This is a confusing statement that isn’t really highlighted in the results. Given that this is a major result of the paper, can it be stated more clearly in the results section?
Reply 18: Yes, we agree with your comments. We added a description in the results section(Ln 221-224).
Comment 19: 368-370 – Same comment as above.
Reply 19: We added more clarity in the results section. (Ln 256-260)
Comment 20: 436-437 – This is the first time that the concept of an “emotional” reaction has been brought up. I’m not sure it belongs here given that there are no questions that actually address this, only whether there is agreement with a particular statement.
Reply 20: Yes, the point that you make is very important, there is really no emotion in our study. We modified the sentence. (Ln 464-467)
Comment 21: 438-439 – Similar argument to above but with men and rationality. I don’t think this is supported by the data.
Reply 21: Yes, we deleted that statement.
Comment 22: 459-462 – This is a sexist statement. I realize that the authors are trying to “prove” that there is value in sustainable harvest, but there is also value in animal rights and ethical animal treatment. If this statement remains in the paper, it should be followed by one that states the ways in which men need to be educated to have greater compassion for animals.
Reply 22: Thank you very much for your advice. We are aware of the possible gender discrimination in similar expressions, although this is not our original intention. We deleted the sentence.
Comment 23: 480-483 – I will admit that I write this review from an American perspective, but I think that it is extremely important to understand that the values and beliefs (and emotions) of the public are equally as relevant as the science when we talk about conservation and management decisions. Scientists need to be better communicators, rather than thinking that the public simply needs to be educated to ensure that correct management decisions are made.
Reply 23: Yes, we agree with you. We have modified the conclusion. (Ln 500-518)

Reviewer 3 Report
The authors present an interesting view into how attitudes in urban China might be shaped by the message as well as the mode of information transmission. Areas that need strengthening include definitions of concepts and terms, as well as greater detail/clarity of the methods. It is unclear how the authors categorize the information being disseminated. How are the authors defining/delimiting utilization, as well as (un)sustainable utilization? And once they define such things, they should explain why they are choosing these descriptions (is it prevailing perspective among practitioners, determined by industry, personal opinion, etc.?) Do the authors equate conservation with utilization? Can the authors speak to the urban rural divide in China with regard to attitudes, reception of information, access to information, etc.? Re: Methods: more information is needed about region selection. The demographic information that was asked for should be outlined in the methods. Need to define tier designations (re: cities). The Methods should also describe how the questions chosen relate to the underlying goals (e.g., promotion of utilization). Also, throughout the manuscript, the authors set up compassionate conservation for strawman argumentation. Regardless of one’s stance on traditional thinking vis a vis alternative thinking in conservation, it is bad form (and poor scholarship) to do this - and ultimately muddles the authors’ intended points. As highlighted on Line 70 with the use of the conditional ‘therefore’, the authors suggest that scientific practitioners by definition are always correct. Do the authors intend this? If the public disagrees with the will of practitioners, what does this mean about the exchange of knowledge and ideas? Is messaging and information dissemination about strategy alone? Should “experts” be open to influence by public sentiment/attitudes/knowledge? What are the intended purpose/goals of "useful" conservation-related messaging?
Line 17: Change trigger to triggers
Line 20: utilization of…
Line 31: Change undoubtably to undoubtedly
Line 59: Recent debate? Reference [1] is from 2002.
Line 67: Attitudes about wildlife trade.
Lines 73/74: How are the authors distinguishing nature and environmental conservation?
Line 74: Typos
Lines 81-84: Need to provide further justification for this claim. The phrasing suggests that the public is hearing only one perspective. And Compassionate Conservation would certainly not be the dominant voice given the influence of more traditional conservation views world wide.
Lines 120-124: Awkward phrasing
Line 150/151/153: Typos
Line 189: Perhaps should rephrase to state that the majority of respondents did not report being vegetarian or religious.
Author Response
General:
The authors present an interesting view into how attitudes in urban China might be shaped by the message as well as the mode of information transmission. Areas that need strengthening include definitions of concepts and terms, as well as greater detail/clarity of the methods.
Reply: Thank you for your valuable suggestion. According to your comments, we have made key modifications to the Introduction and Method section of the article.
Specific:
Comment 1: It is unclear how the authors categorize the information being disseminated. How are the authors defining/delimiting utilization, as well as (un)sustainable utilization? And once they define such things, they should explain why they are choosing these descriptions (is it prevailing perspective among practitioners, determined by industry, personal opinion, etc.?) Do the authors equate conservation with utilization?
Reply 1: Thanks for your suggestion. We have added a definition of sustainable utilization of wildlife in the introduction (Ln 62-68). The classification of information was divided into four categories according to the level of support for the wildlife utilization contained in the information. Do not support the wildlife utilization was divided into not support unsustainable use and do not support all use; Support for use was divided into support for sustainable use and promotion of unsustainable use. The third category is "Support sustainable utilization" instead of "Support unsustainable utilization" which we miswrote and have been corrected. We don't equate conservation with exploitation. There is a complex relationship between conservation and utilization. Appropriate utilization may promote conservation, while excessive utilization is definitely not conducive to conservation.
Comment 2: Can the authors speak to the urban rural divide in China with regard to attitudes, reception of information, access to information, etc.?
Reply 2: The gap between urban and rural areas is a point worth studying. But our study did not reflect that. In our questionnaire, the respondents were not classified as rural or urban residents. We will pay attention to this in future research.
Comment 3: Methods: more information is needed about region selection.
Reply 3: Thank you very much for your advice. We mended the sentence to make it clearer (Ln 137-143).
Comment 4: The demographic information that was asked for should be outlined in the methods. Need to define tier designations (re: cities). The Methods should also describe how the questions chosen relate to the underlying goals (e.g., promotion of utilization).
Reply 4: Yes, we agree with your proposal. The required demographic information is referred to in the methods (Ln 174-176). We add the basis of city stratification in method 2.3 (Ln 184-185). We have supplemented the interpretation of information classification in the Method (Ln 158-165).
Comment 5: Also, throughout the manuscript, the authors set up compassionate conservation for strawman argumentation. Regardless of one’s stance on traditional thinking vis a vis alternative thinking in conservation, it is bad form (and poor scholarship) to do this - and ultimately muddles the authors’ intended points. As highlighted on Line 70 with the use of the conditional ‘therefore’, the authors suggest that scientific practitioners by definition are always correct. Do the authors intend this? If the public disagrees with the will of practitioners, what does this mean about the exchange of knowledge and ideas? Is messaging and information dissemination about strategy alone? Should “experts” be open to influence by public sentiment/attitudes/knowledge? What are the intended purpose/goals of "useful" conservation-related messaging?
Reply 5: Thank you very much for your suggestion. We deleted the sentence.
Comment 6: Line 17: Change trigger to triggers
Reply 6: Thank you, we revised this sentence.
Comment 7: Line 20: utilization of…
Reply 7: We have modified it.
Comment 8: Line 31: Change undoubtably to undoubtedly
Reply 8: Yes, we have modified it.
Comment 9: Line 59: Recent debate? Reference [1] is from 2002.
Reply 9: Thanks for your reminding, we have modified this sentence (Ln 61-62).
Comment 10: Line 67: Attitudes about wildlife trade.
Reply 10: Sorry, we made a mistake in this sentence by missing a comma, which we have corrected (Ln 78).
Comment 11: Lines 73/74: How are the authors distinguishing nature and environmental conservation?
Reply 11: Thanks for your reminding, we deleted "nature conservation".
Comment 12: Typos
Reply 12: We have modified it.
Comment 13: Lines 81-84: Need to provide further justification for this claim. The phrasing suggests that the public is hearing only one perspective. And Compassionate Conservation would certainly not be the dominant voice given the influence of more traditional conservation views worldwide.
Reply 13: Thanks for your suggestion, we have revised this sentence (Ln 89-93).
Comment 14: Lines 120-124: Awkward phrasing 2
Reply 14: Thanks for your comments, we've rewritten that sentence (Ln 129-134).
Comment 15: Line 150/151/153: Typos
Reply 15: We have modified them. (Ln 158-165).
Comment 16: Line 189: Perhaps should rephrase to state that the majority of respondents did not report being vegetarian or religious.
Reply 16: Thanks for your suggestion, we have revised this sentence. (Ln 205).
